# Corpus Augmentation for Neural Machine Translation with Chinese-Japanese Parallel Corpora

**Jinyi Zhang [1,]\* and Tadahiro Matsumoto [2]**

[1] Electronics and Information Systems Engineering Division, Graduate School of Engineering, Gifu University, Gifu 501-1193, Japan

[2] Department of Electrical, Electronic and Computer Engineering, Gifu University, Gifu 501-1193, Japan; tad@gifu-u.ac.jp

\* Correspondence: zhang-jinyi@outlook.com

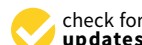

**Featured Application: Machine translation is a subfield of artificial intelligence that investigates transformation of text in the source language into its equivalent in the target language. Neural Machine Translation (NMT) is a recently-proposed framework for translation application based on sequence-to-sequence models: a large neural network is used to translate the source language sequence into the target language sequence. After years of development, NMT has produced richer translation results than ever over various language pairs, becoming a new machine translation model with great potential. As described in this paper, we present a corpus augmentation method. The method has two variations: one is for all language pairs and the other for the Chinese-Japanese language pair. The method generates pseudo-parallel sentence pairs to extend the original parallel corpus. This paper describes the results obtained in the Japanese-Chinese and Chinese-Japanese translation with the ASPEC-JC (Asian Scientific Paper Excerpt Corpus, Japanese-Chinese) corpus, which substantially improved the translation performance. We also supply code that can reproduce our proposed method.**

**Abstract:** The translation quality of Neural Machine Translation (NMT) systems depends strongly on the training data size. Sufficient amounts of parallel data are, however, not available for many language pairs. This paper presents a corpus augmentation method, which has two variations: one is for all language pairs, and the other is for the Chinese-Japanese language pair. The method uses both source and target sentences of the existing parallel corpus and generates multiple pseudo-parallel sentence pairs from a long parallel sentence pair containing punctuation marks as follows: (1) split the sentence pair into parallel partial sentences; (2) back-translate the target partial sentences; and (3) replace each partial sentence in the source sentence with the back-translated target partial sentence to generate pseudo-source sentences. The word alignment information, which is used to determine the split points, is modified with "shared Chinese character rates" in segments of the sentence pairs. The experiment results of the Japanese-Chinese and Chinese-Japanese translation with ASPEC-JC (Asian Scientific Paper Excerpt Corpus, Japanese-Chinese) show that the method substantially improves translation performance. We also supply the code (see Supplementary Materials) that can reproduce our proposed method.

**Keywords:** back translation; Chinese-Japanese translation; corpus augmentation; decoder; encoder; Japanese-Chinese translation; LSTM; neural machine translation; sentence segmentation

---

## 1. Introduction

In recent years, Neural Machine Translation (NMT) has made remarkable achievements [1]. Actually, NMT has achieved good results with large-scale parallel corpora. However, in low-resource languages or domain-defined translation tasks, the parallel corpus is small. Accordingly, the translation performance is severely constrained [2]. Communication during a crisis must be multilingual, and multilingual crisis communications is enabled through translation; in addition, low-resource languages need to be translated [3]. Therefore, studies of NMT under the condition of a low-resource language corpus have high practical value.

Firat et al. [4] used attention mechanisms in several language pairs for NMT and improved the translation performance of other low-resource language pairs by transferring parameters from the used attention mechanisms of the trained NMT models. This method greatly improved the translation performance of low-resource language pairs. However, all experiments were in European language pairs. Language pairs that have markedly different linguistic structures were not included.

Zero-shot translation is a translation mechanism that uses a single NMT engine to translate between multiple languages, even such low-resource languages for which no direct parallel data were provided during training. This type of multi-language method for NMT was mainly proposed by Google [5]. Lakew et al. [6] proposed a simple iterative training procedure that leverages a duality of translations directly generated by the system for the zero-shot directions. Mattoni et al. [7] focused on languages with sparse training data.

The use of a pivot language for low-resource language pairs can also be effective for NMT. For example, given three languages A, B, and C, if no direct parallel corpora exist between A and C, but parallel corpora between A and B and between B and C do exist, then B is useful as a pivot language to achieve translation from A to C. Based on this work, the "teacher–student" framework [8], maximum likelihood estimation method Zheng et al. [9], and joint training method Cheng et al. [10] have also been proposed. Expanding the size of the training data (parallel corpus) is also an effective way to improve the translation performance for NMT in low-resource language pairs. Sennrich et al. [11] proposed a method that generates pseudo-parallel data with monolingual data and back-translation.

In this paper, we propose a method to augment a parallel corpus by sentence segmentation and synthesis. This method has two variations: one is for all language pairs, and the other is for the Chinese-Japanese language pair. Our method splits long sentence pairs (properly speaking, sentence pairs that contain punctuation marks) in the corpus into parallel partial sentence pairs, back-translates the target partial sentences, then synthesizes pseudo-parallel sentence pairs by combining the source partial sentences and the back-translated target partial sentences to expand the corpus size. The method uses the word alignment information of each parallel sentence pair to produce parallel partial sentence pairs. One variation of the method modifies the alignment information with common Chinese character information of the Chinese and Japanese segments. The Chinese language is written with traditional or simplified Chinese characters. In this paper, we consider only simplified Chinese.

In our experiments, we used Luong's NMT system as the base system [12]. It follows an encoder–decoder architecture with global attention. In our case, we chose character-level NMT as the baseline, also because character-level NMT between Japanese and Chinese has better translation performance than word-level NMT.

The main contributions of this paper are the following. We show that we can improve the NMT system's translation performance by mixing generated pseudo-parallel sentence pairs into training data with no monolingual data and without changing the neural network architecture. This process makes our approach applicable to different NMT architectures.

In the remainder of this article, Section 2 presents the related work of this paper. Section 3 gives a brief explanation of the architecture of the NMT and ASPEC-JCcorpus that we are using as the base system. Section 4 describes the proposed method of how to segment long sentences into partial sentences of the corpus and to generate pseudo-parallel sentence pairs to augment the original corpus. Section 5 reports the experimental framework and the results obtained from Japanese-Chinese and Chinese-Japanese translation experiments (with ASPEC-JC [13]) that improve the translation performance. Finally, Section 6 concludes with the discussion of the contributions of this paper.

## 2. Related Work

Several methods have been proposed to expand parallel corpora so far.

The parallel corpus can be constructed quickly using the back-translation method with monolingual target data [11]. Sennrich et al. [14] showed that even simply duplicating the monolingual target data and using them as the source data were sufficient to realize some benefits. The pseudo-parallel corpus can be constructed using this copy method; i.e., the target language sentences are copied as the corresponding source language sentences [15], demonstrating that even poor translations can be beneficial. Data augmentation for low-frequency words has also been proven an effective method [16].

For back-translation methods, the idea of back-translation dates back to statistical machine translation, where it has been used for semi-supervised learning [17]. Gwinnup et al. [18] implemented their NMT system by iteratively applying back-translation. Lample et al. [19] explored the use of generated back-translated data, aided by denoising with a language model trained on the target side. The translation performance can also be improved by iterative back-translation in both high-resource and low-resource scenarios [20]. A more refined idea of back-translation is the dual learning approach of He et al. [21], which integrates training on parallel data and training on monolingual data via round-tripping.

The work of Park et al. [22] presented an analysis of models trained only with synthetic data. In their work, they trained NMT models with parallel corpora composed of (1) synthetic data in the source-side only, (2) synthetic data in the target-side only, and (3) a mixture of parallel sentences of which either the source-side or the target-side was synthetic.

Karakanta et al. [23] used back-translated data to improve MT for low-resource languages. They took advantage of the similarities between a high-resource language and a low-resource language in order to transform the high-resource language data into data similar to the low-resource language using transliteration. The transliteration models were trained on transliteration pairs extracted from Wikipedia article titles. Then, they automatically back-translated monolingual low-resource language data with the models trained on the transliterated high-resource language data and used the resulting parallel corpus to train their final models.

## 3. Neural Machine Translation and ASPEC-JC Corpus

### 3.1. Neural Machine Translation

For this research, we follow the NMT architecture by Luong et al. [12], which implements as a global attentional encoder–decoder neural network with Long Short-Term Memory (LSTM). However, it is noteworthy that our proposed method is not specific to this architecture.

The encoder is a bi-directional recurrent neural network with LSTM units that reads an input sequence $x = (x_1, \ldots, x_m)$ and calculates a forward sequence of hidden states $(\overrightarrow{h}_1, \ldots, \overrightarrow{h}_m)$ and a backward sequence $(\overleftarrow{h}_1, \ldots, \overleftarrow{h}_m)$. The hidden states $\overrightarrow{h}_j$ and $\overleftarrow{h}_j$ are concatenated to obtain annotation vector $h_j$.

The decoder is a recurrent neural network with LSTM units. It predicts a target sequence $y = (y_1, \ldots, y_n)$. Every word (or character in the case of character-level NMT) $y_i$ is predicted based

on a recurrent hidden state $s_i$, the previously predicted word (or character) $y_{i-1}$, and a context vector $c_i$. Here, $c_i$ is computed as the weighted sum of the annotations $h_j$. Finally, the weight of each annotation $h_j$ is computed through an alignment (or attention) model $\alpha_{ij}$, which models the probability that $y_i$ is aligned to $x_j$.

The NMT architecture was used at the character level in our experiments. The Chinese language was written using (simplified or traditional) Chinese characters, which are basically logograms. The Japanese writing system also uses kanji (adopted Chinese characters) along with syllabic kana. Several thousand Chinese characters are in regular use in both languages. Therefore, training models for those languages at the character level could be done more effectively than with a few million kinds of words at the word level. In addition, because the parameters to be trained are far fewer than at the word level, the training and translation time of experiment can probably be kept shorter.

### 3.2. ASPEC-JC Corpus

We conducted experiments with the ASPEC-JC corpus, which was constructed by manually translating Japanese scientific papers into Chinese [13]. The Japanese scientific papers are either the property of JST (Japan Science and Technology Agency) or stored in J-STAGE (Japan's largest electronic journal platform for academic societies).

ASPEC-JC is comprised of four parts: training data (672,315 sentence pairs), development data (2090 sentence pairs), development-test data (2148 sentence pairs), and test data (2107 sentence pairs) with the assumption that it would be used for machine translation research. ASPEC-JC includes both abstracts and some parts of the body texts.

We chose ASPEC-JC as the low-resource corpus compared with other language pairs such as English-French, which usually comprise millions of parallel sentences; the ASPEC-JC corpus has about 672k sentences. We randomly extracted 300k sentence pairs from the training data for experiments.

## 4. Corpus Augmentation by Sentence Segmentation

Sennrich et al. [11] proposed a method to expand a parallel corpus by back-translating target language sentences in monolingual corpora to obtain pseudo-source sentences; the pseudo-source sentences together with the original target sentences were then added to the parallel corpus.

Our method expands the existing parallel corpus with itself, not with any monolingual data, not like some back-translation methods with monolingual data [11,15,16]. Moreover, our method could be combined with other corpus augmentation methods. Our augmentation process includes the following phases: (1) splitting "long" parallel sentence pairs of the corpus into parallel partial sentence pairs, (2) back-translating the target partial sentences, and (3) constructing parallel sentence pairs by combining the source and the back-translated target partial sentences. To be precise, a "long" sentence above means a sentence that contains more than one punctuation mark.

### 4.1. Generating Parallel Partial Sentences

The following procedure generates parallel partial sentence pairs from long parallel sentence pairs.

1.  Obtain the word alignment information from tokenized Japanese-Chinese parallel sentences.
2.  Split the long parallel sentences into segments at the punctuation symbols, such as ",", ";", and ":". Figure 1 presents an example of word alignment information and the segments of a sentence pair.
3.  Obtain source-target segment alignments: For each source segment s-seg$_i$ and target segment t-seg$_j$, count the words in s-seg$_i$ that correspond to the words in t-seg$_j$ according to the word alignment information. The numerical values on the arrows in Figure 2 represent the rate of the correspondence

relation between the segments. We infer that s-seg$_i$ corresponds to t-seg$_j$ if the rate is greater than or equal to a threshold value $\theta_1$.

4. Obtain target-source segment alignments: According to the procedure in 3.
5. Concatenate multiple segments to form a one-to-one relation if there is a one-to-many or many-to-many relation between the segments.

In Figure 2, each sentence is divided into three segments. Two parallel partial sentences are generated.

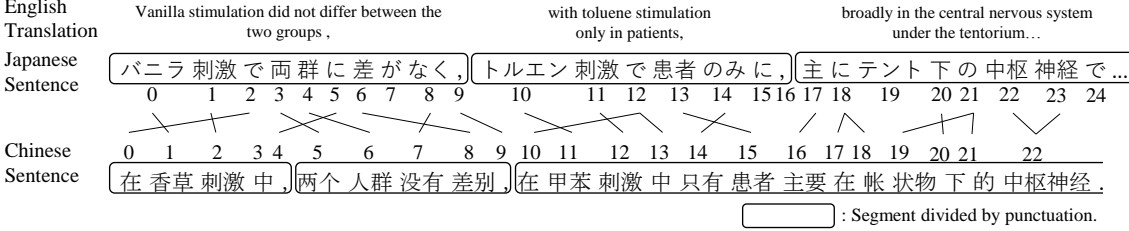

**Figure 1.** Example of word alignment information and sentence segments by punctuation marks.

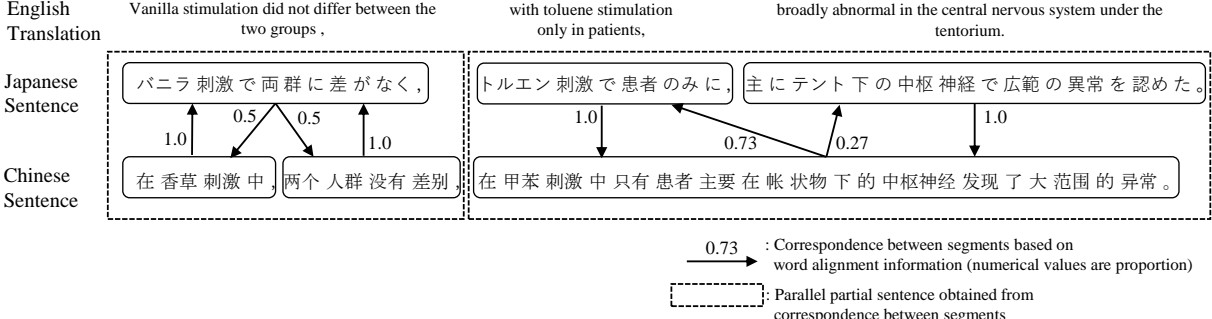

**Figure 2.** Correspondence rates between segments and parallel partial sentences derived from the rates.

### 4.2. Correcting Segments' Correspondence Information Using Common Chinese Characters

Word alignment errors can cause the above procedure to split sentences wrongly. Some of the errors can be avoided by considering the correspondence between Chinese characters in Japanese and Chinese sentences. For that reason, we improved Step 3 of the procedure described above to correct the correspondence rates of segments, as described below.

After World War II, China and Japan respectively simplified their use of Chinese characters in different ways, although Taiwan and Hong Kong still use the traditional Chinese characters. In Unicode, the same character code is assigned to similar Chinese characters. For example, the Japanese kanji 写 (copy) and the simplified Chinese hanzi 写 (write) are unified to U+5199. By contrast, different character codes are assigned to the following pairs, which were originally the same characters: 見 (see) and 见 (see), 発 (depart) and 发 (send out), 広 (wide) and 广 (wide). As a result, one cannot say whether they were originally the same characters based on their codes.

Chu et al. [24] produced a mapping table between Japanese kanji, simplified Chinese hanzi, and traditional Chinese characters. They showed that the table can improve accuracies of word alignment and example-based machine translation between Japanese and Chinese [25]. We replaced Japanese kanji in the parallel data with simplified Chinese hanzi employing Chu's mapping table to calculate the rates of common Chinese characters among segments and used them for correcting correspondence information between segments.

The rate of common Chinese character $\sigma$ for a Japanese and Chinese segment pair is defined as shown below.

$$\sigma = \frac{2n_s}{n_j + n_c} \tag{1}$$

Therein, $n_j$ and $n_c$ respectively denote the number of Chinese characters in the Japanese and Chinese segments, and $n_s$ is the number of the common (shared) Chinese characters. Using the common character rate $\sigma$, we updated the correspondence ratio $\rho$ between segments as:

$$\rho' = \begin{cases} \rho + \sigma \cdot w & (\sigma \geq \theta_2) \\ \rho & (\sigma < \theta_2) \end{cases} \tag{2}$$

where $\theta_2$ and $w$ respectively stand for a threshold value and a weight ($\rho'$ is no longer a ratio).

An example of the correction is presented in Figure 3. In this figure, the correspondence of partial sentences including "電流滴定法" (amperometric titration method) are corrected in the lower part.

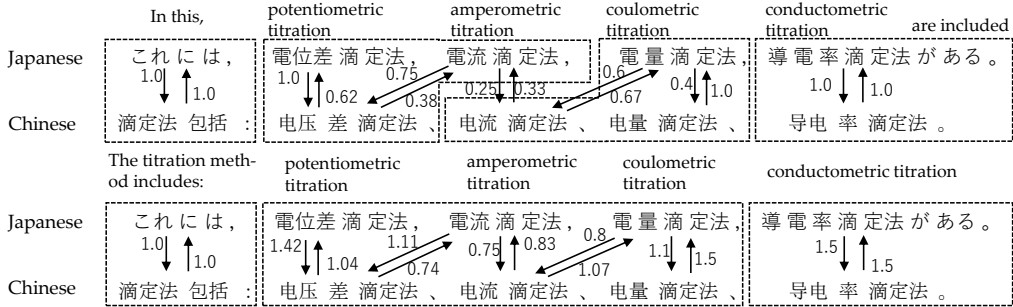

**Figure 3.** Correction of segments' correspondence information using common Chinese character information: upper, before correction; lower, after correction.

### 4.3. Corpus Augmentation by Generated Parallel Partial Sentences

Using the parallel partial sentences generated according to the procedure described in Section 4.1, pseudo-parallel sentences were constructed using the following procedure. If some segment occurs that does not correspond to any other segment or a correspondence relation (arrows in Figure 2) is crossing, then we do not use the sentence pair because the partial sentences of the sentence are probably not parallel.

1. Back-translate the target partial sentences into the source language with a translation model built from parallel data.
2. For each sentence, create a pseudo-source sentence that is partly different from the original source sentence by replacing a part of the original sentence with a partial sentence obtained using back-translation. As a result, it is possible to generate the same number of variations of pseudo-source language sentences as the number of partial sentences. For example, if a sentence is divided into two partial sentences, two pseudo-source sentences will be created. Table 1 shows the pseudo-source language sentences generated from the Japanese sentence of Figure 2.
3. Copy the target sentences corresponding to the created pseudo-source sentences to produce pseudo-parallel sentences.
4. Add the generated pseudo-parallel sentences to the original parallel corpus.

**Table 1.** Examples of an original source sentence and pseudo-source sentences with English translations. "//" denotes the splitting position.

| Original/Generated Sentences | Input Japanese Sentence | English Translation of the Input Japanese Sentence |
| --- | --- | --- |
| Source sentence (original) | バニラ刺激で両群に差がなく，// トルエン刺激で患者のみに，主にテント下の中枢神経で広範の異常を認めた。 | Vanilla stimulation did not differ between the two groups, // with toluene stimulation only in patients, broadly abnormal in the central nervous system under the tentorium. |
| Pseudo-source sentence 1 (pseudo- and original) | 香草刺激では，両群に差はなかった // トルエン刺激で患者のみに，主にテント下の中枢神経で広範の異常を認めた。 | There was no difference between the two groups with vanilla stimulation // only in patients, toluene stimulation showed extensive abnormalities, mainly in the central nervous system under the tent. |
| Pseudo-source sentence 2 (original and pseudo-) | バニラ刺激で両群に差がなく，// トルエン刺激には主に帳票物下の中枢神経で広範囲の異常が認められた。 | With vanilla stimulation there was no difference between both groups, // a wide range of abnormality was confirmed mainly in the central nervous system under the slap for toluene stimulation. |

## *4.4. Use of Sentences Not Divided into Partial Sentences*

Some sentence pairs could not be divided into partial sentence pairs through the procedure in Section 4.3, even though they included punctuation marks. We retried generating pseudo-parallel sentences using only the target sentences of those undivided sentence pairs as follows.

1. Extract target sentences from the undivided sentence pairs that can be split into multiple segments.
2. Back-translate the extracted target language sentences into the source language.
3. Split the target language sentence and its back-translation result into segments in the manner used for Step 2 in Section 4.1.
4. Let $t$ and $\bar{t}$ be respectively a target sentence and its back-translation result; let $n$ and $m$ be respectively the numbers of the segments of $t$ and $\bar{t}$; i.e., $t = (s_1, s_2, \ldots, s_n)$ and $\bar{t} = (s'_1, s'_2, \ldots, s'_n)$. Extract sentence pairs $(t, \bar{t})$ such that $(n = m)$ and $(n \geq 2)$.

   (4-1) Back-translate each segment $s_i$ of $t$ to $\overline{s_i}$.
   (4-2) For $i$ $(1 \leq i \leq n)$, replace $s'_i$ in $\bar{t}$ with $\overline{s_i}$ to generate $n$ pseudo-source sentences: $\bar{t}^{(1)} = (\overline{s_1}, s'_2, \ldots, s'_n), \bar{t}^{(2)} = (s'_1, \overline{s_2}, \ldots, s'_n), \ldots, \bar{t}^{(n)} = (s'_1, s'_2, \ldots, \overline{s_n})$.
   (4-3) Make $n$ pseudo-parallel sentence pairs $(\bar{t}^{(1)}, t), \ldots, (\bar{t}^{(n)}, t)$ from each obtained pseudo-source sentence $\bar{t}^{(i)}$ and the target sentence $t$.

We also supply code (see Supplementary Materials) that can reproduce our proposed method.

## 5. Evaluation and Translation Results

### *5.1. Experiment Settings*

We followed the NMT architecture by Luong et al. [12] and implemented the NMT architecture using OpenNMT [26]. The LSTM model had one layer, each with 512 cells, with embedding size of 512. The parameters were uniformly initialized in $(-0.1, 0.1)$, using plain SGD, starting with a learning rate of 1.0 until Epoch 6, and subsequently 0.5 times for each epoch. The max-batch size was 100. The normalized gradient was rescaled whenever its norm exceeded one. Because of the amounts of training data (150k and 300k as the baseline) were relatively small, the dropout probability was set as 0.5 to avoid overfitting. Decoding was performed by beam search with a beam size of five. The maximum length of a sentence was 250 by default, but it was set to 500 because it became much longer in character-level MT.

Because sentences in Japanese and Chinese are written without spaces, we tokenized them with MeCab http://taku910.github.io/mecab for Japanese and Jieba http://github.com/fxsjy/jieba for Chinese. We employed fast_align to obtain word alignment information, which was symmetrized using the included atool command http://github.com/clab/fast_align.

BiLingual Evaluation Understudy (BLEU) is an algorithm for evaluating the quality of text that has been machine-translated from one natural language to another [27]. Translation Error Rate (TER) is an error metric for machine translation that measures the number of edits required to change a system output into one of the references [28]. The BLEU and TER scores were calculated on the same test data (2109 sentence pairs) and the same development-test data (2148 sentence pairs) of the ASPEC-JC corpus for each method, using the "tools/score.lua" of OpenNMT after word segmentation. In other words, we took the word-level evaluation. The validation perplexity (i.e., perplexity with the same development data for each method) usually stopped declining around Epoch 10 in the settings above. The average of BLEU and TER scores from that point to Epoch 16 was taken as the BLEU and TER values.

*5.2. Selection of Thresholds $\theta_1$, $\theta_2$ and Weight $w$*

Following, we discuss the selection of thresholds $\theta_1$, $\theta_2$ and weight $w$ described in Section 4.

To determine the threshold $\theta_1$, we have conducted an experiment to investigate the changes in the number of generated partial sentence pairs with 300k sentences. As shown in Figure 4, we found that the number of the generated partial sentence pairs had a peak when $\theta_1$ was 0.5.

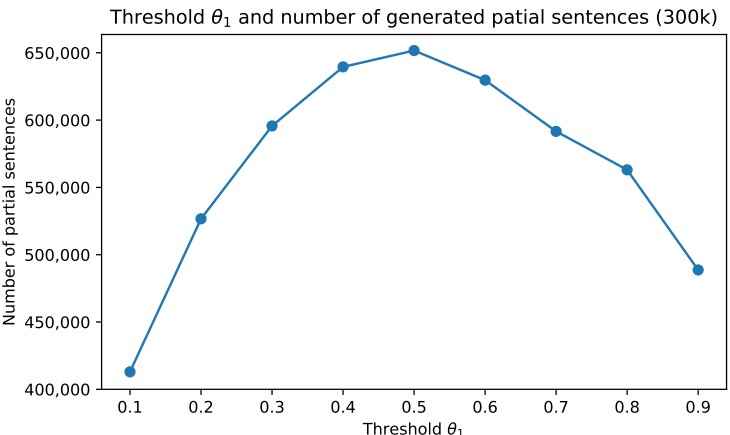

**Figure 4.** Threshold $\theta_1$ and number of generated partial sentences (300k).

We also did the experiments for the threshold $\theta_2$ and the weight $w$ and manually evaluated and observed the error rate of the aligned sentences, with 300k sentences as shown in Table 2.

**Table 2.** Experiment results of the error rate of aligned sentences with the threshold $\theta_2$ and the weight $w$ for 300k training data. "Without cc" indicates without using the common Chinese character information; "With cc" indicates using the common Chinese character information.

| The Error Rate of Aligned Sentences (%) | $\theta_2 = 0.3$ | | $\theta_2 = 0.5$ | | $\theta_2 = 0.7$ | |
|---|---|---|---|---|---|---|
| | Without cc | With cc | Without cc | With cc | Without cc | With cc |
| $w = 0.3$ | 8.8 | 6.8 | 7.6 | 5.6 | 9.2 | 9.0 |
| $w = 0.5$ | 2.1 | 1.7 | 1.7 | 0.8 | 3.3 | 2.2 |
| $w = 0.7$ | 7.0 | 3.0 | 6.0 | 3.8 | 7.6 | 7.2 |

In order to determine the threshold $\theta_2$ and the weight $w$, we manually checked the alignment of the generated partial sentence pairs with 9000 (= 500 × 18) partial sentence pairs randomly extracted from the 300k sentences. Table 2 shows the error rates. To confirm the effectiveness of the common Chinese character information, we compared the error rates with and without using the information. The error rate of aligned sentences had the lowest value when the weight $\theta_2 = w = 0.5$ with using the common Chinese character information. The common Chinese character information was helpful for reducing the error rate of aligned sentences.

Due to the above results, for the experiments described later, we set $\theta_1 = 0.5$ for the Proposed 1 method and $\theta_2 = w = 0.5$ for the Proposed 2 method.

*5.3. Experiment Results and Discussion*

Table 3 shows the information of the ASPEC-JC corpus. We randomly extracted 300k sentence pairs from 672k training data of the ASPEC-JC corpus for experiments as the training data.

**Table 3.** The information of the ASPEC-JC corpus.

| ASPEC-JC Corpus | Number of Sentence Pairs |
|---|---|
| Training data | 672,315 |
| Development (dev) data | 2090 |
| Development-test (dev-test) data | 2148 |
| Test data | 2107 |

Figure 5 shows the changes of BLEU scores on the same test data by epochs with 300k original training data. Figure 6 shows the changes of TER scores on the same test data by epochs with 300k original training data. Figure 7 shows changes of the validation perplexity values on the same development data by epochs with 300k original training data. The proposed methods obtained BLEU and TER scores better than the other methods on the test data in both cases.

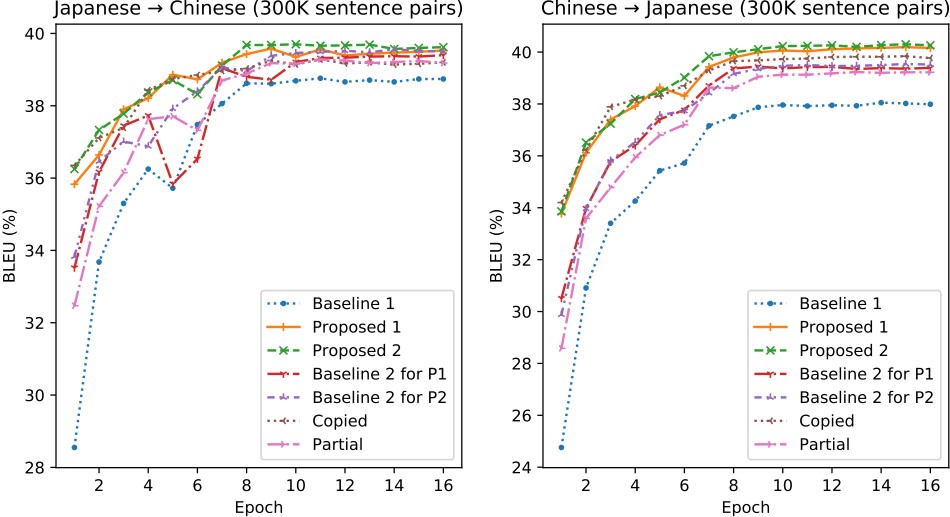

**Figure 5.** BiLingual Evaluation Understudy (BLEU) scores on the test data of 300k sentences. "P1" denotes the Proposed 1 method, "P2" denotes the Proposed 2 method.

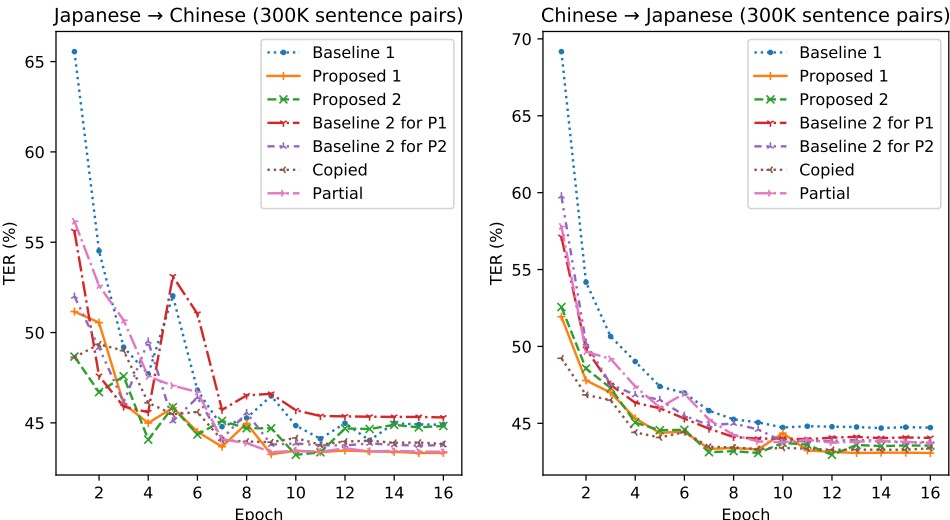

**Figure 6.** Translation Error Rate (TER) scores on the test data of 300k sentences. "P1" denotes the Proposed 1 method, "P2" denotes the Proposed 2 method.

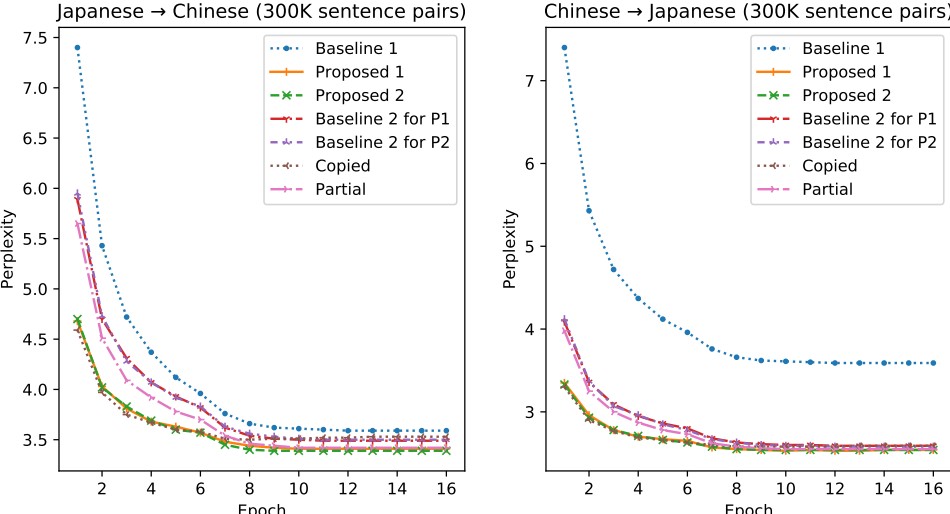

**Figure 7.** Validation perplexity values on the development data of 300k sentences. "P1" denotes the Proposed 1 method, "P2" denotes the Proposed 2 method.

The best (lowest) perplexities on the development (dev) data after they stopped declining and the BLUE and TER scores of the translation results on the same test-data and the same development-test data by each method are presented in Tables 4 and 5.

**Table 4.** Experiment results of Japanese→Chinese Machine Translation (MT) with 300k training data. "ppl" denotes perplexity. "Dev" denotes development data. "Dev-test" denotes development-test data.

| Method | Chinese→Japanese | | | | | | | |
|---|---|---|---|---|---|---|---|---|
| | # Sentences | | # Back-Translated | ppl | BLEU (%) | | TER (%) | |
| | Raw | Used | | Dev | Dev-Test | Test | Dev-Test | Test |
| Baseline 1 | 300k | 300k | 0 | 3.6 | 38.5 | 38.7 | 44.0 | 44.8 |
| Baseline 2 for P1 | 518k | 518k | 218k | 3.5 | 39.1 | 39.4 | 43.9 | 45.3 |
| Baseline 2 for P2 | 531k | 531k | 231k | 3.5 | 39.2 | 39.5 | 43.0 | 43.8 |
| Copied | 977k | 977k | 0 | 3.5 | 38.9 | 39.2 | 43.7 | 43.9 |
| Partial | 984k | 984k | 0 | 3.5 | 38.9 | 39.2 | 43.0 | 43.4 |
| Proposed 1 | 952k | 923k | 218k | 3.4 | 39.2 | 39.5 | 44.2 | 43.4 |
| Proposed 2 | 977k | 945k | 231k | 3.4 | 39.3 | 39.7 | 42.9 | 43.4 |

**Table 5.** Experiment results of Chinese→Japanese MT with 300k training data. "ppl" denotes perplexity. "Dev" denotes development data. "Dev-test" denotes development-test data.

| Method | Chinese→Japanese | | | | | | | |
|---|---|---|---|---|---|---|---|---|
| | # Sentences | | # Back-Translated | ppl | BLEU (%) | | TER (%) | |
| | Raw | Used | | Dev | Dev-Test | Test | Dev-Test | Test |
| Baseline 1 | 300k | 300k | 0 | 2.7 | 38.1 | 38.0 | 44.9 | 44.8 |
| Baseline 2 for P1 | 518k | 518k | 218k | 2.6 | 40.0 | 39.4 | 44.5 | 44.1 |
| Baseline 2 for P2 | 529k | 529k | 229k | 2.6 | 40.1 | 39.5 | 43.6 | 43.8 |
| Copied | 972k | 972k | 0 | 2.6 | 39.7 | 39.8 | 43.5 | 43.3 |
| Partial | 984k | 984k | 0 | 2.6 | 39.0 | 39.2 | 43.9 | 43.8 |
| Proposed 1 | 952k | 947k | 218k | 2.5 | 40.2 | 40.1 | 42.9 | 43.3 |
| Proposed 2 | 972k | 967k | 229k | 2.5 | 40.5 | 40.2 | 43.6 | 43.4 |

"Baseline 1" was a character-level translation with the 300k original training data. The back-translation models for corpus augmentation were constructed using the 300k original training data of "Baseline 1".

"Proposed 1" was the proposed data augmentation method without consideration of the common Chinese character rates and reuse of the undivided sentences (Section 4.4), which can be used for all language pairs. This method expanded the parallel corpus from the original 300k sentence pairs to 952k sentence pairs in both directions (Japanese→Chinese and Chinese→Japanese). Two hundred and eighteen thousand sentence pairs from 300k training data were used for back-translation in both directions.

"Proposed 2" was the proposed method, which considered the common Chinese character rates and reused undivided sentences, which can be used only for Chinese-Japanese language pairs. This method expanded the parallel corpus from 300k sentence pairs to 977k and 972k sentence pairs in Japanese→Chinese and Chinese→Japanese directions, respectively. Two hundred twenty seven thousands and 218k sentence pairs from 300k original training data were used for back-translation, in each direction.

"Baseline 2 for P1" was the back-translation method that back-translated the same data as the Proposed 1 (P1) method did (218k from original training data). The experiment of this method aimed to compare Proposed 1 with the back-translation method (Baseline 2) on the same back-translated data. "Baseline 2 for P2" was the back-translation method that back-translated the same data as the Proposed 2 (P2) method did (231k from original training data). The experiment of this method aimed to compare Proposed 2 with the back-translation method (Baseline 2) on the same back-translated data.

"Copied" was the method that added duplicate copies of the training data as the same times as the Proposed 2 method did. The experiment of this method aimed to highlight differences between the

generated pseudo-parallel sentences pairs and unchanged sentences pairs. This method expanded the parallel corpus from 300k sentence pairs to 977k and 972k sentence pairs in each direction.

"Partial" was the method that augmented the corpus with parallel partial sentences generated by the procedure in Section 4.3, without back-translating and mixing the partial sentences. The experiment of this method aimed to confirm that the mixing step (Section 4.3, Step 2) was necessary. This method expanded the parallel corpus from 300k sentence pairs to 984k sentence pairs in both directions.

"# sentences" in the tables denotes the size (the number of sentence pairs) of training data, whereas "# back-translated" denotes the number of parallel sentence pairs used for back-translation processing, i.e., the corpus augmentation, in each method. "ppl" denotes the best (lowest) perplexity values on the development (dev) data in each method.

In the case of 300k training data, the number of parallel sentence pairs augmented by Proposed 2 was 977k in the Japanese→Chinese direction. However, only 945k pairs were used as the training data. This is because the translation errors in the back-translation steps sometimes generated unusually long pseudo-source sentences where the same words or phrases occurred repeatedly in a sentence, and such sentence pairs were removed due to exceeding the upper limit of the training data length (500 characters). As a result, the used training data size of Proposed 2 was 32k (3.3%) smaller than that of "Copied" (977k). For this reason, the training sentence numbers of "Copied" and "Proposed 2" in Tables 4 and 5 are different. Hence, we added the columns "Raw" and "Used" in the tables to denote the numbers of generated sentences (raw data before removing) and used sentences (after removing unusually long sentences), respectively.

The proposed methods obtained BLEU and TER scores better than the baselines did on the development-test data and test data in both cases. Although there were translation errors and unnatural expressions in the generated pseudo-source sentences, the BLEU scores were higher and the TER scores were lower than "Copied" and "Partial" on the development-test data and test data in both directions, Japanese→Chinese and Chinese→Japanese. The BLEU scores of the "Partial" method were lower than the proposed methods, in both directions. Therefore, the mixing step (Section 4.3, Step 2) was necessary. These results demonstrate that the proposed methods were effective for augmenting small-scale parallel corpora to improve translation performance for Japanese→Chinese and Chinese→Japanese NMT.

Comparing "Proposed 1", "Proposed 2" with "Baseline 2 for P1", "Baseline 2 for P2" in the tables, the results of the proposed methods were nearly identical and better : the proposed method was effective at improving translation accuracy in both directions, Japanese→Chinese and Chinese→Japanese.

The experiments described above (Tables 4 and 5) proved the effectiveness of the proposed methods. Nevertheless, our approach was based on only the original parallel data and did not require any additional monolingual data, unlike the back-translation method of Sennrich et al. [11]. Most methods of corpus augmentation were applied to pair monolingual training data with automatic back-translation and then treat them as additional parallel training data. Therefore, we added comparison experiments.

We conducted comparison experiments using 150k and 300k sentences that were randomly extracted from 672k training data of ASPEC-JC as the original data and used the remaining 522k and 372k sentences as the monolingual data.

For the comparison experiment, we only implemented our "Proposed 2" method because the experiments described above proved that the "Proposed 2" method was better than the "Proposed 1" method in most cases with the Chinese-Japanese parallel corpus. Translation results obtained on the test data and development-test data are shown in Tables 6–9 with 150k and 300k original training data in both directions, respectively.

**Table 6.** Experiment results of Japanese→Chinese MT with 150k sentences and 522k monolingual sentences. "ppl" denotes perplexity. "Dev" denotes development data. "Dev-test" denotes development-test data.

| Method | Chinese→Japanese | | | | | | | |
|---|---|---|---|---|---|---|---|---|
| | # Sentences | | # Back-Translated | ppl | BLEU (%) | | TER (%) | |
| | Raw | Used | | Dev | Dev-Test | Test | Dev-Test | Test |
| Baseline 1 | 150k | 150k | 0 | 4.3 | 36.5 | 36.5 | 48.5 | 50.1 |
| Baseline 2 + mono (522k) | 672k | 672k | 522k | 3.8 | 38.8 | 39.1 | 44.6 | 44.7 |
| 150k + mono (522k) + P2 | 2313k | 2201k | 525k | 3.7 | 38.9 | 39.1 | 43.9 | 44.4 |

**Table 7.** Experiment results of Chinese→Japanese MT with 150k sentences and 522k monolingual sentences. "ppl" denotes perplexity. "Dev" denotes development data. "Dev-test" denotes development-test data.

| Method | Chinese→Japanese | | | | | | | |
|---|---|---|---|---|---|---|---|---|
| | # Sentences | | # Back-Translated | ppl | BLEU (%) | | TER (%) | |
| | Raw | Used | | Dev | Dev-Test | Test | Dev-Test | Test |
| Baseline 1 | 150k | 150k | 0 | 3.1 | 35.4 | 35.5 | 48.4 | 47.5 |
| Baseline 2 + mono (522k) | 672k | 672k | 522k | 2.8 | 39.3 | 39.1 | 45.1 | 44.6 |
| 150k + mono (522k) + P2 | 2239k | 2134k | 515k | 2.7 | 40.6 | 40.1 | 43.7 | 43.7 |

**Table 8.** Experiment results of Japanese→Chinese MT with 300k sentences and 372k monolingual sentences. "ppl" denotes perplexity. "Dev" denotes development data. "Dev-test" denotes development-test data.

| Method | Chinese→Japanese | | | | | | | |
|---|---|---|---|---|---|---|---|---|
| | # Sentences | | # Back-Translated | ppl | BLEU (%) | | TER (%) | |
| | Raw | Used | | Dev | Dev-Test | Test | Dev-Test | Test |
| Baseline 1 | 300k | 300k | 0 | 3.6 | 38.5 | 38.7 | 44.0 | 44.8 |
| Baseline 2 + mono (372k) | 672k | 672k | 372k | 3.4 | 39.6 | 39.7 | 42.8 | 44.2 |
| 300k + mono (372k) + P2 | 2287k | 2234k | 522k | 3.4 | 39.8 | 40.1 | 42.6 | 43.3 |

**Table 9.** Experiment results of Chinese→Japanese MT with 300k sentences and 372k monolingual sentences. "ppl" denotes perplexity. "Dev" denotes development data. "Dev-test" denotes development-test data.

| Method | Chinese→Japanese | | | | | | | |
|---|---|---|---|---|---|---|---|---|
| | # Sentences | | # Back-Translated | ppl | BLEU (%) | | TER (%) | |
| | Raw | Used | | Dev | Dev-Test | Test | Dev-Test | Test |
| Baseline 1 | 300k | 300k | 0 | 2.7 | 38.1 | 38.0 | 44.9 | 44.8 |
| Baseline 2 + mono (372k) | 672k | 672k | 372k | 2.6 | 40.5 | 39.9 | 43.3 | 43.3 |
| 300k + mono (372k) + P2 | 2223k | 2213k | 501k | 2.5 | 41.8 | 41.4 | 42.2 | 42.5 |

For Tables 6 and 7, "Baseline 1" is a character-level translation, which did not process anything with 150k original training data. The back-translation models were constructed using 150k original training data of Baseline 1 for each method, before corpus augmentation. The back-translation models were constructed using 150k original training data of "Baseline 1" for each method, before corpus augmentation. "Baseline 2 + mono (522k)" was the back-translation method of Sennrich et al. [11], which back-translated the remaining 522k target language sentences of 672k training data to generate 522k pseudo-source sentences directly with no segmentation; the 522k pseudo-source sentences, together with the 522k target sentences,

expanded the parallel corpus from 150k sentence pairs to 672k sentence pairs. The experiment of this method aimed to confirm the effectiveness of applying our proposed methods to the augmented data by the back-translation method of Sennrich et al. [11].

"150k + mono (522k) + P2" represents the combination method of "Baseline 2 + mono (522k)" (672k training data) and "Proposed 2". "150k + mono (522k) + P2" back-translated 525k and 515k from the "Baseline 2 + mono (522k)" (672k training data), so that the numbers of sentence pairs were increased from 672k to 2313k and 2239k in both directions.

For Tables 8 and 9, "Baseline 1" is a character-level translation which did not process anything with 300k original training data. The back-translation models were constructed using 300k original training data of Baseline 1 for each method, before corpus augmentation. The back-translation models were constructed using 300k original training data of "Baseline 1" for each method, before corpus augmentation. "Baseline 2 + mono (372k)" was the back-translation method of Sennrich et al. [11], which back-translated the remaining 372k target language sentences of 672k training data to generate 372k pseudo-source sentences directly with no segmentation; the 372k pseudo-source sentences, together with the 372k target sentences, expanded the parallel corpus from 300k sentence pairs to 672k sentence pairs. The experiment of this method aimed to confirm the effectiveness of applying our proposed methods to the augmented data by the back-translation method of Sennrich et al. [11].

"300k + mono (372k) + P2" represents the combination method of "Baseline 2 + mono (372k)" (672k training data) and "Proposed 2". "300k + mono (372k) + P2" back-translated 522k and 501k from the "Baseline 2 + mono (372k)" (672k training data), so that the numbers of sentence pairs were increased from 672k to 2287k and 2223k in both directions.

The proposed methods obtained nearly identical and better results of BLEU and TER scores than in the case of baseline methods on the development-test data and test data. These comparison experiments demonstrate that our proposed method can augment the extended data by the other corpus augmentation methods to yield better translation performance. In the future, we plan to combine the proposed methods with other augmentation approaches, as our results suggest it may be more beneficial than only back-translation.

The salient benefits of the proposed method are that it requires no monolingual data and that, without changing the neural network architecture, our method can generate more pseudo-parallel sentences. Moreover, it can be combined with other augmentation methods.

## 6. Conclusions and Future Work

In this paper, we proposed simple, but effective approaches to augment corpora of NMT for all language pairs and for Chinese-Japanese language pairs, by segmenting long sentences in the parallel corpus, using back-translation and generating pseudo-parallel sentences pairs. We demonstrated that the approaches engender more pseudo-parallel sentences. Consequently, they obtained equal or higher translation performance than in the case of the back-translation method for NMT with the ASPEC-JC corpus. We have also reported improvements over the baseline systems.

In the future, we plan to combine the proposed methods with other augmentation approaches, as our results suggest it may be more beneficial than only back-translation. On the other hand, as there is only one pseudo-partial sentence in each generated pseudo-sentence, we should consider various combinations of pseudo-partial sentences to generate more pseudo-sentences.

**Supplementary Materials:** The following are available online at http://www.mdpi.com/2076-3417/9/10/2036/s1.

**Author Contributions:** J.Z. and T.M. conceived of and designed the methodology and experiments. J.Z. wrote the manuscript. T.M. reviewed and edited the manuscript. All authors read and approved the final manuscript.

**Funding:** This research received no external funding.

**Acknowledgments:** J.Z. was supported by China Scholarship Council (CSC) under CSC Grant No. 201708050078. All authors wish to thank the anonymous reviewers and Editors for their insightful comments and assistance.

**Conflicts of Interest:** The authors declare no conflict of interest, financial or otherwise, in relation to this paper.

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
