# Peer review of "Corpus Augmentation for Neural Machine Translation with Chinese-Japanese Parallel Corpora"

_applsci, doi:10.3390/app9102036_

Round 1
Reviewer 1 Report
The authors present a method to automatically augment training data for MT.
The paper is quite well written, and the methods and ideas clearly exposed.
My main concern is with the potential impact of the proposed approach.
First. Given the small difference in BLEU results, I consider that including confidence intervals or tests about the significance of the differences between different approaches of paramount importance to understand the advantages of the proposed methods. Experiments with additional testsets (even for the same language pair) will also help to generalise the reported results.
Second. Given that the proposed method is an improved back-translation approach. I'd suggest to take back-translation as the baseline for comparison.
Third. Following the previous point, I think a way to highlight the advantages of Proposed 1 (P1) and Proposed 2 (P2) would be to carry out back-translation experiments using the same back-translated data as the P1 and P2 respectivelly. By comparing P1 and P2 against back-translation on the same back-translated data, we should be able to show the highest efficiency of the two proposed methods.
Finally, I'd suggest the authors to tone down a bit their conclusions in section 6. Given the reported results, it is not clear that P1 or P2 are better approaches than standard back-translation.
Author Response
A Summary of Changes
Manuscript Title: Corpus Augmentation for Neural Machine Translation with Chinese-Japanese Parallel Corpora
Authors: Jinyi Zhang, Tadahiro Matsumoto
Review’s response:
Dear Editor and Reviewers:
First of all, we would like to express our sincere gratitude to the editor and the anonymous reviewers for their critical reading and enlightening suggestions for improving this manuscript. Based on the reviewers’ valuable and helpful comments, we have made correction which we hope meet with approval. Revised portions are marked in yellow in the paper. In the following, we will describe the overall changes made in the revision (Please note the reviewers’ comments are written in blue italics) and our point-by-point reply to the reviewers’ comments.
Overall changes
1. The English expressions in the original manuscript have been much improved and enhanced.
2. The improper statements in the original manuscript have been removed or revised.
3. The references were updated.
4. Added experimental results using an additional test set and another metric TER (in addition to BLEU).
5. The conclusion and related work sections have been expanded.
----------------------------------------------
REVIEW NO. 1
Comments to the authors:
The authors present a method to automatically augment training data for MT.
The paper is quite well written, and the methods and ideas clearly exposed.
Response:
Thank you for your positive comments.
My main concern is with the potential impact of the proposed approach.
First. Given the small difference in BLEU results, I consider that including confidence intervals or tests about the significance of the differences between different approaches of paramount importance to understand the advantages of the proposed methods. Experiments with additional test-sets (even for the same language pair) will also help to generalize the reported results.
Response:
Thank you for your suggestions. We added the results of experiments with an additional test set (the development-test data of ASPEC-JC corpus, 2,148 sentence pairs) and used TER as an additional metric.
Second. Given that the proposed method is an improved back-translation approach. I'd suggest to take back-translation as the baseline for comparison.
Response:
We are appreciative of your suggestion. Now, the back-translation method is referred to as “Baseline 2” (the old one is referred to as “Baseline 1”) for comparison in the revised manuscript.
Third. Following the previous point, I think a way to highlight the advantages of Proposed 1 (P1) and Proposed 2 (P2) would be to carry out back-translation experiments using the same back-translated data as the P1 and P2 respectively. By comparing P1 and P2 against back-translation on the same back-translated data, we should be able to show the highest efficiency of the two proposed methods.
Response:
Thank you for your suggestions. We conducted additional experiments with comparing P1 and P2 against back-translation on the same back-translated data.
Added some explanations:
“Proposed 1” is the proposed data augmentation method without consideration of the common Chinese character rates and reuse of the undivided sentences (Section 4.4), which can be used for all language pairs. This method expands the parallel corpus from original 300k sentence pairs to 952k sentence pairs in both directions, Japanese-Chinese and Chinese-Japanese. 218k sentence pairs from 300k training data were used for back-translation in both directions.
“Proposed 2” is the proposed method, which considers the common Chinese character rates and reuses undivided sentences, which can be used only for Chinese–Japanese language pairs. This method expands the parallel corpus from 300k sentence pairs to 977k and 972k sentence pairs in Japanese-Chinese and Chinese-Japanese directions, respectively. 227k and 218k sentence pairs from 300k original training data were used for back-translation in each direction.
“Baseline 2 for P1” is the back-translation method that back-translates the same data as the Proposed 1 method does (218k from original training data). The experiment of this method
aims to compare Proposed 1 with the back-translation method (Baseline 2) on the same back-translated data.
“Baseline 2 for P2” is the back-translation method that back-translates the same data as the Proposed 2 method does (231k from original training data). The experiment of this method aims to compare Proposed 2 with the back-translation method on the same back-translated data.
“Copied” is the method that adds duplicate copies of the training data as the same times as the Proposed 2 method does. The experiment of this method aims to highlight differences between the generated pseudo-parallel sentences pairs and unchanged sentences pairs. This method expands the parallel corpus from 300k sentence pairs to 977k and 972k sentence pairs in each direction.
“Partial” is the method that augments the corpus with parallel partial sentences generated by the procedure in Section 4.3 without back-translating and mixing the partial sentences. The experiment of this method aims to confirm the mixing step (Section 4.3, step 2) is necessary. This method expands the parallel corpus from 300k sentence pairs to 984k sentence pairs in both directions.
“# sentences” in the tables denotes the size (the number of sentence pairs) of training data, whereas “# back-translated” denotes the number of parallel sentence pairs used for back-translation processing, i.e., the corpus augmentation, in each method. “ppl” denotes the best (lowest) perplexity values on the development data in each method.
In the case of 300k training data, the number of the parallel sentence pairs augmented by Proposed 2 is 977k in the Japanese-Chinese direction. However, only 945k pairs were used as the training data. This is because the translation errors in the back-translation steps sometimes generate unusually long pseudo-source sentences where the same words or phrases occur repeatedly in a sentence, and such sentence pairs are removed due to exceeding the upper limit of the training data length (500 characters). As a result, the used training data size of Proposed 2 is 32k (3.3%) smaller than that of “Copied” (977k). For this reason, the training sentence numbers of ‘Copied’ and ‘Proposed 2’ in Table 4 and Table 5 are different. Hence, we add the columns “Raw” and “Used” in the tables to denote the numbers of generated sentences (raw data before removing) and used sentences (after removing unusually long sentences), respectively.
Finally, I'd suggest the authors to tone down a bit their conclusions in section 6. Given the reported results, it is not clear that P1 or P2 are better approaches than standard back-translation.
Response:
Thank you for your suggestion. The Section 6 has been rewrote as below.
In this paper, we proposed simple but effective approaches to augment corpora of NMT for all language pairs and specific Chinese-Japanese language pairs, by segmenting long sentences in the parallel corpus, using back-translation and generating pseudo-parallel sentences pairs. We demonstrated that the approaches engenders more pseudo-parallel sentences. Consequently, it obtains equal or higher translation performance than in the case of back-translation method for NMT with ASPEC-JC corpus. We have also reported improvements over the baseline system.
In the future we plan to combine the proposed methods with other augmentation approaches as our results suggest it may be more beneficial than only back-translation. On the other hand, as there is only one pseudo partial sentence in each generated pseudo sentence, we should consider more various combinations of pseudo partial sentences to generate more pseudo sentences.
We tried our best to improve the manuscript and made some changes in the manuscript. These changes will not influence the content and framework of the paper. And here we did not list the changes but marked in yellow in revised paper.
We appreciate for your warm work earnestly, and hope that the correction will meet with approval.
Once again, thank you very much for your comments and suggestions.

Reviewer 2 Report
The paper presents a novel approach for data augmentation for the purposes of neural machine translation. The approach is defined and applied in the scope of Chinese->Japanese and Japanese->Chinese translation. While it is noted that the approach is applicable for any other language pair, the authors show that using additional language-specific resources to improve the word alignment yields better results.
Positive points:
- Novel, language independent method. Compatible with other methods for data augmentation.
- Good experimental work.
- The paper is quite high level and could be understood by a broader audience.
Negative points:
- Extend on the related work.
- From the beginning make clear whether you are talking about simplified or traditional Chinese.
- With respect to the experiments, it is clear what has been done, but it needs explanation about why are you doing all these experiments. What is their purpose? What point do they prove?
- Conclusion and related work sections are very brief.
Specific comments:
* Featured application/Abstract:
- line 2: "of the source language into the target language" -> "text in the source language into its equivalent in the target language"
- line 24: "shows" -> "show"
* 1. Introduction (starting from line 28):
- I don't understand the relevance of the many citations in the first paragraph. Especially "also useful for the Arabic language translation task [4]"
- If you are talking about relevant work on low-resource translation you need more references - e.g., related to MT in crisis situations (check the Interact project and related literature by Sharon O'Brien); zero-shot MT for low-resource languages, e.g., Mattoni et al 2017, Zero-Shot Translation for Indian Languages with Sparse Data; and the work of FBK, e.g., Lakew et al., Improving Zero-Shot Translation of Low-Resource Languages.
- line 46/47: Google's work on Multilingual translation is focused on NOT using pivot languages but zero-shot translation. From your text it seems that the focus of [8] was on using pivot data.
- line 62: 'the character-level NMT' -> 'character-level NMT' in both occurrences.
- line 63: 'the word-level NMT' -> 'word-level NMT'
- line 68: 'report' -> 'article'
* 2. Related work:
- line 82/83: 'In addition, the data augmentation for the low-frequency words is an effective method [17].' -> 'Data augmentation for low-frequency words has also been proven an effective method [17].'
- line 84/85: Reformulate the sentence. It doesn't sound good.
* 3. Neural Machine Translation and ASPEC-JC Corpus
* 3.1.
- Reformulate or remove the first paragraph. It is not well structured.
- line 96: '... [13], which we describe briefly here. This NMT system is implemented as a global...' -> '...[13], which implements a global...'.
- line 98: Remove the sentence 'Then we simply ... level.'. It is not the correct place for this sentence. You can mention this information in the section where you talk about the vocabulary of the networks.
- lines 100-104: move to the end of the section and reformulate
- line 105: 'bi-directional neural network' -> 'bi-directional recurrent neural network'
- you give only one equation. What is the point of it? By its own it doesn't contribute to the paper. You need to either put more equations to show how source and target sequences relate and, potentially, link it to your work, or just remove the equation.
* 3.2.
- What is the point of the development-test data? How do you use it?
- line 126-130: The paragraph is confusing and not clear.
* 4. Corpus Augmentation by Sentence Segmentation.
The introductory part needs to be made more clear regarding what is that you do and how does it differ to other methods (i.e. Sennrich et al. [12])
* 4.1.
- line 150: Label 3. as 'Obtain source-target segment aligments' and describe the procedure as it is now following the label. Then 4. (line 155) should look like '4. Obtain target-source segment alignments: According to the procedure in 3.'
- line 156: '5. The multiple side segments are concatenated to be a one...' -> '5. Concatenate multiple segments to form a one...'
- line 160 seems like it should become a new subsection.
* 4.2.
- line 187: '...in the previous section,...' -> '...according to the procedure described in Section 4.1....'
- line 189: 'or correspondence relation' -> 'or a/the correspondence relation'
- Table 1: needs to have one more row for all the three different versions.
- Table 1: add column labels.
- line 193: '2. Create a ...' -> '2. For each sentence, create a ...'
- line 197: '..., two pseudo-source sentences will be created.' -> '..., maximum two pseudo-source sentences will be created.'
* 4.3.
- line 210: 4. is not clear. Also 5, 6 and 7 seem to me need to be as subitems of 4. Make sure these are very well and clearly explained. These are essential for understanding your contribution.
* 5. Evaluation and Translation Results
* 5.1.
- line 219. First you say in Section 3 that you are following th architecture of Luong et al. and here you say that you implemented your NMT system using OpenNMT. Can you make it clear how are you constructing your NMT models?
- line 223: Why dropout of 0.5? Typically it is something like 0.2.
- paragraph between lines 229-234: Are computing BLEU on the test or the dev set? A suggestion - use also TER at least. Having only BLEU, especially for NMT, is not that indicative of quality.
- line 235: Which training data do you use for your baseline. At least make a reference to Table 3 to let people know what data you are using for the baseline system.
* 5.2.
- line 239: 'In this subsection, we will discuss ... and weight w described in Section 4.' -> 'Following, we discuss the selection of ... and the weight w, described in Section 4.'
- line 244: 'We also have done' -> 'We also did'
- line 244: 'manually observed' is not a good term. 'manually evaluated and observed...' maybe a good substitute.
- line 246: The term 'parallelism' is not the best in this case. I think you should use 'alignment' here.
You need to reformulate these paragraphs.
- Title of Table 2. 'The previous value... information.' You can say 'the first value' and then 'the second value'. But I suggest you split these values in different columns - with a common label regarding the \theta values and separate label for each of the columns. And that for each column with two values.
* 5.3.
- line 256: which is the test data? Use a table and for each model identify what is the training data, what is the dev data and what is the test data. Then add times and perplexity.
- line 259: 'it can be used for all language pairs'. This statement is very important. You should make it clear in the beginning of your paper that that is the case. It is part of your contribution.
- line 261/262. If 'Copied' is the method that adds duplicate copies of the training data as the same times as the 'Proposed 2' method, why are then the number of sentences in Table 3 and Table 4 different between 'Copied' and 'Proposed 2'.
- line 262: 'This method will show...' -> 'This method aims to...'
- line 264: 'is the method that back-translates' -> 'is the system that back-translates'
- line 266: 'that augment the corpus' -> 'that augments the corpus'
- line 267: 'with the parallel partial sentences' -> 'with parallel partial sentences'
- The last two sentences in this paragraph (line 268-270) need to go in the discussion section.
- line 274/275: The information in this sentence should be more detailed. Say how many do you first detect. Then how many do you copy. Then you can put this sentence.
- line 279/280: 'higher than the baseline in both cases of 300k training data.' -> 'higher than the baseline, trained on 300k sentence pairs, in both cases.
- line 291: '...monolingual data. Most...' -> '... monolingual data, unlike back-translation. Most...'.
- line 302: 'than in case of the back-translation' -> 'than in the case of back-translation'
- line 304/305: The last sentence: 'It is useful to ...augmentation methods' -> 'In the future we plan to combine the proposed methods with other augmentation approaches as our results suggest it may be more beneficial than only back-translation.'
- line 308: 'although back-translation is very time consuming'. I disagree with this statement. I think your method could be equally time consuming or even more. If you have a model for back-translation then it is a matter of translating the monolingual data you have. Check the work of Poncellas et al. 2018 ("Investigating Backtranslation in Neural Machine Translation") for more information and details.
* 6. Conclusion
I would suggest to rename this section to 'Conclusions and Future Work'
You need to expend it and add future work that you may think of doing. Your current idea has potential and so you can do more work on it.
Author Response
A Summary of Changes
Manuscript Title: Corpus Augmentation for Neural Machine Translation with Chinese-Japanese Parallel Corpora
Authors: Jinyi Zhang, Tadahiro Matsumoto
Review’s response:
Dear Editor and Reviewers:
First of all, we would like to express our sincere gratitude to the editor and the anonymous reviewers for their critical reading and enlightening suggestions for improving this manuscript. Based on the reviewers’ valuable and helpful comments, we have made correction which we hope meet with approval. Revised portions are marked in yellow in the paper. In the following, we will describe the overall changes made in the revision (Please note the reviewers’ comments are written in blue italics) and our point-by-point reply to the reviewers’ comments.
Overall changes
1. The English expressions in the original manuscript have been much improved and enhanced.
2. The improper statements in the original manuscript have been removed or revised.
3. The references were updated.
4. Added experiments and computed another score metrics(TER) in the revised manuscript.
5. The conclusion and related work sections have been expanded.
----------------------------------------------
REVIEW NO. 2
Comments to the authors:
The paper presents a novel approach for data augmentation for the purposes of neural machine translation. The approach is defined and applied in the scope of Chinese->Japanese and Japanese->Chinese translation. While it is noted that the approach is applicable for any other language pair, the authors show that using additional language-specific resources to improve the word alignment yields better results.
Positive points:
- Novel, language independent method. Compatible with other methods for data augmentation.
- Good experimental work.
- The paper is quite high level and could be understood by a broader audience.
Response:
Thank you very much for your positive comments.
Negative points:
- Extend on the related work.
- From the beginning make clear whether you are talking about simplified or traditional Chinese.
- With respect to the experiments, it is clear what has been done, but it needs explanation about why are you doing all these experiments. What is their purpose? What point do they prove?
- Conclusion and related work sections are very brief.
Response:
Thank you very much for your helpful and constructive suggestions. Some explanations have been added in the revised manuscript.
1. We extended the related work section as follows:
Related Work
Several methods have been proposed to expand parallel corpora so far.
The parallel corpus can be constructed quickly using the back-translation method with monolingual target data [11]. Sennrich et al. [14] showed that even simply duplicating the monolingual target data and using them as the source data was sufficient to realize some benefits. The pseudo-parallel corpus can be constructed using this copy method; i.e., the target language sentences are copied as the corresponding source language sentences [15], demonstrating that even poor translations can be beneficial. Data augmentation for low-frequency words has also been proven an effective method [16].
For back-translation methods, the idea of back-translation dates back to statistical machine translation, where it has been used for semi-supervised learning [17]. Gwinnup et al. [18] implemented their NMT system with iteratively applying back-translation. Lample et al. [19] explored the use of generated back-translated data, aided by denoising with a language model trained on the target side. The translation performance can also be improved by iterative back-translation in both high-resource and low-resource scenarios [20]. A more refined idea of back-translation is the dual learning approach of He et al. [21], which integrates training on parallel data and training on monolingual data via round-tripping.
The work of Park et al. [22] presented an analysis of models trained only with synthetic data. In their work, they trained NMT models with parallel corpora composed of: 1) synthetic data in the source-side only; 2) synthetic data in the target-side only; and 3) a mixture of parallel sentences of which either the source-side or the target-side is synthetic.
Karakanta et al. [23] used back-translated data to improve MT for low-resource languages. They took advantage of the similarities between the high-resource language and the low-resource language in order to transform the high-resource language data into data similar to the low-resource language using transliteration. The transliteration models were trained on transliteration pairs extracted from Wikipedia article titles. Then, they automatically back-translated monolingual low-resource language data with the models trained on the transliterated high-resource language data and used the resulting parallel corpus to train their final models.
The following references were added:
[17] Bojar, O.; Tamchyna, A. Improving Translation Model by Monolingual Data. Proceedings of the Sixth Workshop on Statistical Machine Translation; Association for Computational Linguistics: Stroudsburg, PA, USA, 2011; pp. 330–336.
[20] Poncelas, A.; 437 Shterionov, D.; Way, A.; de Buy Wenniger, G.M.; Passban, P. Investigating Backtranslation in Neural Machine Translation. CoRR 2018, abs/1804.06189, [1804.06189].
[21] He, D.; Xia, Y.; Qin, T.; Wang, L.; Yu, N.; Liu, T.Y.; Ma, W.Y. Dual Learning for Machine Translation. Advances in Neural Information Processing Systems 29; Lee, D.D.; Sugiyama, M.; Luxburg, U.V.; Guyon, I.; Garnett, R., Eds. Curran Associates, Inc., 2016, pp. 820–828.
[22] Park, J.; Song, J.; Yoon, S. Building a Neural Machine Translation System Using Only Synthetic Parallel Data. CoRR 2017, abs/1704.00253.
[23] Karakanta, A.; Dehdari, J.; van Genabith, J. Neural machine translation for low-resource languages without parallel corpora. Machine Translation 2018, 32, 167–189. doi:10.1007/s10590-017-9203-5.
2. We added explanations that this research was done with simplified Chinese.
3. We added explanations that purposes of the experiments (including additional experiments) and the points they proved in Section 5 as follows:
“Proposed 1” is the proposed data augmentation method without consideration of the common Chinese character rates and reuse of the undivided sentences (Section 4.4), which can be used for all language pairs. This method expands the parallel corpus from original 300k sentence pairs to 952k sentence pairs in both directions, Japanese-Chinese and Chinese-Japanese. 218k sentence pairs from 300k training data were used for back-translation in both directions.
“Proposed 2” is the proposed method, which considers the common Chinese character rates and reuses undivided sentences, which can be used only for Chinese–Japanese language pairs. This method expands the parallel corpus from 300k sentence pairs to 977k and 972k sentence pairs in Japanese-Chinese and Chinese-Japanese directions, respectively. 227k and 218k sentence pairs from 300k original training data were used for back-translation in each direction.
“Baseline 2 for P1” is the back-translation method that back-translates the same data as the Proposed 1 method does (218k from original training data). The experiment of this method aims to compare Proposed 1 with the back-translation method (Baseline 2) on the same back-translated data.
“Baseline 2 for P2” is the back-translation method that back-translates the same data as the Proposed 2 method does (231k from original training data). The experiment of this method aims to compare Proposed 2 with the back-translation method on the same back-translated data.
“Copied” is the method that adds duplicate copies of the training data as the same times as the Proposed 2 method does. The experiment of this method aims to highlight differences between the generated pseudo-parallel sentences pairs and unchanged sentences pairs. This method expands the parallel corpus from 300k sentence pairs to 977k and 972k sentence pairs in each direction.
“Partial” is the method that augments the corpus with parallel partial sentences generated by the procedure in Section 4.3 without back-translating and mixing the partial sentences. The experiment of this method aims to confirm the mixing step (Section 4.3, step 2) is necessary. This method expands the parallel corpus from 300k sentence pairs to 984k sentence pairs in both directions.
......
The experiments described above (Table 4 and Table 5) prove the effectiveness of the proposed methods. Nevertheless, our approach is based on only the original parallel data and does not require any additional monolingual data, unlike back-translation method of Sennrich et al. [11]. Most methods of corpus augmentation are applied to pair monolingual training data with automatic back-translation and then treat them as additional parallel training data. Therefore, we have added comparison experiments.
......
“Baseline 2+mono (522k)” is the back-translation method of Sennrich et al. [11], which back-translates the remaining 522k target language sentences of 672k training data to generate 522k pseudo-source sentences directly with no segmentation; the 522k pseudo-source sentences, together with the 522k target sentences, expand the parallel corpus from 150k sentence pairs to 672k sentence pairs. The experiment of this method aims to confirm the effectiveness of applying our proposed method to the augmented data by the back-translation method of Sennrich et al. [11].
......
“Baseline 2+mono (372k)” is the back-translation method of Sennrich et al. [11], which back-translates the remaining 372k target language sentences of 672k training data to generate 372k pseudo-source sentences directly with no segmentation; the 372k pseudo-source sentences, together with the 372k target sentences, expand the parallel corpus from 300k sentence pairs to 672k sentence pairs. The experiment of this method aims to confirm the effectiveness of applying our proposed method to the augmented data by the back-translation method of Sennrich et al. [11].
4. The conclusion and related work sections have been expanded.
Specific comments:
* Featured application/Abstract:
- line 2: "of the source language into the target language" -> "text in the source language into its equivalent in the target language"
- line 24: "shows" -> "show"
Response:
Thank you for your advice. We have revised them according to your kind advice.
* 1. Introduction (starting from line 28):
- I don't understand the relevance of the many citations in the first paragraph. Especially "also useful for the Arabic language translation task [4]"
Response:
Thank you for your comment. We have removed the sentences including citations [2], [3], [4] and [5] in the first paragraph of Introduction.
- If you are talking about relevant work on low-resource translation you need more references - e.g., related to MT in crisis situations (check the Interact project and related literature by Sharon O'Brien); zero-shot MT for low-resource languages, e.g., Mattoni et al 2017, Zero-Shot Translation for Indian Languages with Sparse Data; and the work of FBK, e.g., Lakew et al., Improving Zero-Shot Translation of Low-Resource Languages.
- line 46/47: Google's work on Multilingual translation is focused on NOT using pivot languages but zero-shot translation. From your text it seems that the focus of [8] was on using pivot data.
Response:
Thank you for your suggestions. We have added some relevant works of zero-shot MT in the introduction, and Google's work has moved in zero-shot MT works as follows:
Communication during a crisis must be multilingual and multilingual crisis communications is enabled through translation, and low-resource languages need to be translated [3].
Zero-shot translation is a translation mechanism which uses a single NMT engine to translate between multiple languages, even such low-resource languages for which no direct parallel data was provided during training. This type of multi-language methods for NMT was mainly proposed by Google [5]. Lakew et al. [6] proposed a simple iterative training procedure that leverages a duality of translations directly generated by the system for the zero-shot directions. Mattoni et al. [7] focused on languages with sparse training data.
The following references were added:
[3] Sharon O’Brien, Chao-Hong Liu, Andy Way, João Graça, André MarMns, Helena Moniz, Ellie Kemp, Rebecca Petras. The INTERACT Project and Crisis MT. Proceedings of MT Summit XVI, Vol.2: Users and Translators Track, 2017, pp. 56–76.
[6] Lakew, S.M.; Lotito, Q.; Negri, M.; Turchi, M.; Federico, M. Improving Zero-Shot Translation of Low-Resource Languages. 2018, Vol. abs/1811.01389, [1811.01389].
[7] Mattoni, G.; Nagle, P.; Collantes, C.; Shterionov, D. Zero-Shot Translation for Indian Languages with Sparse Data. MT Summit 2017; , 2017.
- line 62: 'the character-level NMT' -> 'character-level NMT' in both occurrences.
- line 63: 'the word-level NMT' -> 'word-level NMT'
- line 68: 'report' -> 'article'
Response:
Thank you for pointing them out. We have corrected them.
* 2. Related work:
- line 82/83: 'In addition, the data augmentation for the low-frequency words is an effective method [17].' -> 'Data augmentation for low-frequency words has also been proven an effective method [17].'
- line 84/85: Reformulate the sentence. It doesn't sound good.
Response:
Thank you for your advice. We have revised the lines 82-83 according to your kind advice.
For line 84/85, we revised the sentence ‘Gwinnup et al.[18] explained iteratively applying back-translation in their system, but did not report successful experiments.’ to ‘Gwinnup et al.[18] implemented their NMT system with iteratively applying back-translation.’
* 3. Neural Machine Translation and ASPEC-JC Corpus
* 3.1.
- Reformulate or remove the first paragraph. It is not well structured.
- line 96: '... [13], which we describe briefly here. This NMT system is implemented as a global...' -> '...[13], which implements a global...'.
- line 98: Remove the sentence 'Then we simply ... level.'. It is not the correct place for this sentence. You can mention this information in the section where you talk about the vocabulary of the networks.
- lines 100-104: move to the end of the section and reformulate
- line 105: 'bi-directional neural network' -> 'bi-directional recurrent neural network'
- you give only one equation. What is the point of it? By its own it doesn't contribute to the paper. You need to either put more equations to show how source and target sequences relate and, potentially, link it to your work, or just remove the equation.
Response:
Thank you for your suggestions. We have carefully revised the original manuscript.
1. We removed the first paragraph in Section 3.1.
2. line 96 was corrected as ‘[13], which implements a global...'.
3. (line 98) we removed the sentence 'Then we simply ... level.'.
4. lines 100-104: were moved to the end of the section and be reformulated as follows:
The NMT architecture is used at character level in our experiments. The Chinese language is written using (simplified or traditional) Chinese characters, which are basically logograms. The Japanese writing system also uses kanji (adopted Chinese characters) along with syllabic kana. Several thousand Chinese characters are in regular use in both languages. Therefore, training models for those languages at character-level could be done more effectively than with a few million kinds of words at word-level. In addition, because the parameters to be trained are far fewer than at word-level, the training and translation time of experiment can probably be kept shorter.
5. line 105 was corrected as 'bi-directional recurrent neural network'.
6. We removed the equation.
* 3.2.
- What is the point of the development-test data? How do you use it?
- line 126-130: The paragraph is confusing and not clear.
Response:
Thank you for your comments.
1. We did not use the development-test (devtest) data of ASPEC-JC, but we have conducted additional experiments with the development-test data and added their results to the manuscript. We calculated the TER and BLEU scores on the development-test data.
2. We removed the lines 126-130.
* 4. Corpus Augmentation by Sentence Segmentation.
The introductory part needs to be made more clear regarding what is that you do and how does it differ to other methods (i.e. Sennrich et al. [12])
Response:
We are appreciative of your comment. Now, the back-translation method (Sennrich et al. [12]) is referred to as ‘Baseline 2’ (the old one is referred to as ‘Baseline 1’) for comparison in the revised manuscript.
The difference is rewrote as below:
Our method expands the existing parallel corpus with itself, not with any monolingual data, not like some back-translation methods with monolingual data [12][16][17]. Moreover, our method could be combined with other corpus augmentation methods.
* 4.1.
- line 150: Label 3. as 'Obtain source-target segment alignments' and describe the procedure as it is now following the label. Then 4. (line 155) should look like '4. Obtain target-source segment alignments: According to the procedure in 3.'
- line 156: '5. The multiple side segments are concatenated to be a one...' -> '5. Concatenate multiple segments to form a one...'
- line 160 seems like it should become a new subsection.
Response:
Thank you for your suggestions. We have revised the sentences according to your kind suggestions.
* 4.2.
- line 187: '...in the previous section,...' -> '...according to the procedure described in Section 4.1....'
- line 189: 'or correspondence relation' -> 'or a/the correspondence relation'
- Table 1: needs to have one more row for all the three different versions.
- Table 1: add column labels.
- line 193: '2. Create a ...' -> '2. For each sentence, create a ...'
- line 197: '..., two pseudo-source sentences will be created.' -> '..., maximum two pseudo-source sentences will be created.'
Response:
Thank you for your advice. We have carefully revised the sentences according to your kind advice. -
* 4.3.
- line 210: 4. is not clear. Also 5, 6 and 7 seem to me need to be as subitems of 4. Make sure these are very well and clearly explained. These are essential for understanding your contribution.
Response:
Thank you for your comment. We have revised the steps 4-7.
* 5. Evaluation and Translation Results
* 5.1.
- line 219. First you say in Section 3 that you are following the architecture of Luong et al. and here you say that you implemented your NMT system using OpenNMT. Can you make it clear how are you constructing your NMT models?
Response:
Thank you very much for your helpful and constructive suggestion. Some explanations has been added in the revised manuscript as shown below:
“We follow the NMT architecture by Luong et al. [12] and implement the NMT architecture using OpenNMT [26].”
- line 223: Why dropout of 0.5? Typically it is something like 0.2.
Response:
Thank you for your question. We added the explanations:
Because of the amounts of training data (150k and 300k as the baseline) is relatively small, we set dropout probability of 0.5 to avoid overfitting.
The following papers also set dropout probability of 0.5 in their experiments:
1. “Improving Neural Machine Translation Models with Monolingual Data, Rico Sennrich and Barry Haddow and Alexandra Birch, ACL2016”
2. “Multi-Channel Encoder for Neural Machine Translation, Hao Xiong, Zhongjun He, Xiaoguang Hu, Hua Wu, AAAI2018”
3. “A Stochastic Decoder for Neural Machine Translation, Philip Schulz, Wilker Aziz, Trevor Cohn, ACL2018”
- paragraph between lines 229-234: Are computing BLEU on the test or the dev set? A suggestion - use also TER at least. Having only BLEU, especially for NMT, is not that indicative of quality.
Response:
Thank you for your suggestion. We used TER in addition to BLEU on the test data (2,109 sentence pairs) and development-test data (2,148 sentence pairs) of ASPEC-JC corpus.
We also added the explanations:
Translation Error Rate (TER) is an error metric for machine translation that measures the number of edits required to change a system output into one of the references[28] .
The BLEU and TER scores were calculated on the same test data (2,109 sentence pairs) and the same development-test data (2,148 sentence pairs) of ASPEC-JC corpus for each method.
and a reference:
[28] Snover, M.; Dorr, B.; Schwartz, R.; Micciulla, L.; Makhoul, J. A study of translation edit rate with targeted human annotation. In Proceedings of Association for Machine Translation in the Americas, 2006, pp.454 223–231.
- line 235: Which training data do you use for your baseline. At least make a reference to Table 3 to let people know what data you are using for the baseline system.
Response:
Thank you for your advice. We moved these explanations to Section 5.3 to let people know what data are used for each experiment. The explanations are revised as shown below:
We randomly extracted 300k sentence pairs from 672k training data of ASPEC-JC as the training data for our experiments.
…
“Baseline 1” is a character-level translation with the 300k original training data. The back-translation models for corpus augmentation are constructed using the 300k original training data of “Baseline 1”.
* 5.2.
- line 239: 'In this subsection, we will discuss ... and weight w described in Section 4.' -> 'Following, we discuss the selection of ... and the weight w, described in Section 4.'
- line 244: 'We also have done' -> 'We also did'
- line 244: 'manually observed' is not a good term. 'manually evaluated and observed...' maybe a good substitute.
- line 246: The term 'parallelism' is not the best in this case. I think you should use 'alignment' here.
You need to reformulate these paragraphs.
Response:
Thank you for your suggestions. We have carefully revised the original manuscript by your kind suggestions.
- Title of Table 2. 'The previous value... information.' You can say 'the first value' and then 'the second value'. But I suggest you split these values in different columns - with a common label regarding the \theta values and separate label for each of the columns. And that for each column with two values.
Response:
Thank you for your advice. Table 2 has been updated.
* 5.3.
- line 256: which is the test data? Use a table and for each model identify what is the training data, what is the dev data and what is the test data. Then add times and perplexity.
Response:
Thank you for your suggestion. We added some explanations to let people know what data are used for each experiment. We also added Table 3 and perplexity results on the development data with 300k sentence pairs.
Some explanations we added for Table 4 and 5 are shown as below:
“Proposed 1” is the proposed data augmentation method without consideration of the common Chinese character rates and reuse of the undivided sentences (Section 4.4), which can be used for all language pairs. This method expands the parallel corpus from original 300k sentence pairs to 952k sentence pairs in both directions (Japanese-Chinese and Chinese-Japanese). 218k sentence pairs from 300k training data were used for back-translation in both directions.
“Proposed 2” is the proposed method, which considers the common Chinese character rates and reuses undivided sentences, which can be used only for Chinese–Japanese language pairs. This method expands the parallel corpus from 300k sentence pairs to 977k and 972k sentence pairs in Japanese-Chinese and Chinese-Japanese directions, respectively. 227k and 218k sentence pairs from 300k original training data were used for back-translation in each direction.
“Baseline 2 for P1” is the back-translation method that back-translates the same data as the Proposed 1 method does (218k from original training data). The experiment of this method aims to compare Proposed 1 with the back-translation method (Baseline 2) on the same back-translated data.
“Baseline 2 for P2” is the back-translation method that back-translates the same data as the Proposed 2 method does (231k from original training data). The experiment of this method aims to compare Proposed 2 with the back-translation method on the same back-translated data.
“Copied” is the method that adds duplicate copies of the training data as the same times as the Proposed 2 method does. The experiment of this method aims to highlight differences between the generated pseudo-parallel sentences pairs and unchanged sentences pairs. This method expands the parallel corpus from 300k sentence pairs to 977k and 972k sentence pairs in each direction.
“Partial” is the method that augments the corpus with parallel partial sentences generated by the procedure in Section 4.3 without back-translating and mixing the partial sentences. The experiment of this method aims to confirm the mixing step (Section 4.3, step 2) is necessary. This method expands the parallel corpus from 300k sentence pairs to 984k sentence pairs in both directions.
# sentences” in the tables denotes the size (the number of sentence pairs) of training data, whereas “# back-translated” denotes the number of parallel sentence pairs used for back-translation processing, i.e., the corpus augmentation, in each method. “ppl” denotes the best (lowest) perplexity values on the development data in each method.
- line 259: 'it can be used for all language pairs'. This statement is very important. You should make it clear in the beginning of your paper that that is the case. It is part of your contribution.
Response:
Thank you for your suggestions. This statement is revised in the Section 1:
“In this paper, we propose a method to augment a parallel corpus by sentence segmentation and synthesis.” as
“In this paper, we propose a method to augment parallel corpus by sentence segmentation and synthesis. This method has two variations; one is for all language pairs and the other is for the Chinese-Japanese language pair. ”
- line 261/262. If 'Copied' is the method that adds duplicate copies of the training data as the same times as the 'Proposed 2' method, why are then the number of sentences in Table 3 and Table 4 different between 'Copied' and 'Proposed 2'.
Response:
Thank you for your question. We added some explanations:
In the case of 300k training data, the number of the parallel sentence pairs augmented by Proposed 2 is 977k in the Japanese-Chinese direction. However, only 945k pairs were used as the training data. This is because the translation errors in the back-translation steps sometimes generate unusually long pseudo-source sentences where the same words or phrases occur repeatedly in a sentence, and such sentence pairs are removed due to exceeding the upper limit of the training data length (500 characters). As a result, the used training data size of Proposed 2 is 32k (3.3%) smaller than that of “Copied” (977k). For this reason, the training sentence numbers of ‘Copied’ and ‘Proposed 2’ in Table 4 and Table 5 are different. Hence, we add the columns “Raw” and “Used” in the tables to denote the numbers of generated sentences (raw data before removing) and used sentences (after removing unusually long sentences), respectively.
Table 4 and 5 have been updated.
- line 262: 'This method will show...' -> 'This method aims to...'
- line 264: 'is the method that back-translates' -> 'is the system that back-translates'
- line 266: 'that augment the corpus' -> 'that augments the corpus'
- line 267: 'with the parallel partial sentences' -> 'with parallel partial sentences'
- The last two sentences in this paragraph (line 268-270) need to go in the discussion section.
Response:
Thank you for your advice. We have revised the lines according to your kind advice. The last two sentences in this paragraph (line 268-270) were moved to the discussion section.
- line 274/275: The information in this sentence should be more detailed. Say how many do you first detect. Then how many do you copy. Then you can put this sentence.
Response:
Thank you for your suggestions.
For line 274/275, we rewrote “In the case of 300k training data, the numbers of the parallel sentences augmented by Proposed 1 and Proposed 2 are 54k (5.5%) and 32k (3.3%) smaller than the one by “Copied”, respectively.” as
“ In the case of 300k training data, the number of the parallel sentence pairs augmented by Proposed 2 is 977k in the Japanes-Chinese direction. However, only 945k pairs were used as the training data. This is because the translation errors in the back-translation steps sometimes generate unusually long pseudo-source sentences where the same words or phrases occur repeatedly in a sentence, and such sentence pairs are removed due to exceeding the upper limit of the training data length (500 characters). As a result, the used training data size of Proposed 2 is 32k (3.3%) smaller than that of “Copied” (977k).”
- line 279/280: 'higher than the baseline in both cases of 300k training data.' -> 'higher than the baseline, trained on 300k sentence pairs, in both cases.
- line 291: '...monolingual data. Most...' -> '... monolingual data, unlike back-translation. Most...'.
- line 302: 'than in case of the back-translation' -> 'than in the case of back-translation'
- line 304/305: The last sentence: 'It is useful to ...augmentation methods' -> 'In the future we plan to combine the proposed methods with other augmentation approaches as our results suggest it may be more beneficial than only back-translation.'
Response:
Thank you for your advice. We have revised the lines according to your kind advice.
- line 308: 'although back-translation is very time consuming'. I disagree with this statement. I think your method could be equally time consuming or even more. If you have a model for back-translation then it is a matter of translating the monolingual data you have. Check the work of Poncellas et al. 2018 ("Investigating Back-translation in Neural Machine Translation") for more information and details.
Response:
Thank you for your kind suggestions. About line 308: 'although back-translation is very time consuming', we removed this expression to avoid confusing.
* 6. Conclusion
I would suggest to rename this section to 'Conclusions and Future Work'
You need to expand it and add future work that you may think of doing. Your current idea has potential and so you can do more work on it.
Response:
Thank you for your suggestion. The Section 6 has been renamed to 'Conclusions and Future Work' and revised as below:
In this paper, we proposed simple but effective approaches to augment corpora of NMT for all language pairs and specific Chinese-Japanese language pairs by segmenting long sentences in the parallel corpus, using back-translation and generating pseudo-parallel sentences pairs. We demonstrated that the approaches engenders more pseudo-parallel sentences. Consequently, it obtains equal or higher translation performance than in the case of back-translation method for NMT with ASPEC-JC corpus. We have also reported improvements over the baseline systems.
In the future we plan to combine the proposed methods with other augmentation approaches as our results suggest it may be more beneficial than only back-translation. On the other hand, as there is only one pseudo partial sentence in each generated pseudo sentence, we should consider more various combinations of pseudo partial sentences to generate more pseudo sentences.
We tried our best to improve the manuscript and made some changes in the manuscript. These changes will not influence the content and framework of the paper. And here we did not list the changes but marked in yellow in revised paper.
We appreciate for your warm work earnestly, and hope that the correction will meet with approval.
Once again, thank you very much for your comments and suggestions.
